# EZH2-Mediated H3K27 Trimethylation in the Liver of Mice Is an Early Epigenetic Event Induced by High-Fat Diet Exposure

**DOI:** 10.3390/nu16193260

**Published:** 2024-09-26

**Authors:** Giulia Pinton, Mattia Perucca, Valentina Gigliotti, Elena Mantovani, Nausicaa Clemente, Justyna Malecka, Gabriela Chrostek, Giulia Dematteis, Dmitry Lim, Laura Moro, Fausto Chiazza

**Affiliations:** 1Department of Pharmaceutical Sciences, Università del Piemonte Orientale (UPO), Largo Donegani 2, 28100 Novara, Italy; giulia.pinton@uniupo.it (G.P.); valentina.gigliotti@uniupo.it (V.G.); gabriela.chrostek98@gmail.com (G.C.); giulia.dematteis@uniupo.it (G.D.);; 2Department of Health Sciences, Interdisciplinary Research Center of Autoimmune Diseases (IRCAD), Università del Piemonte Orientale (UPO), Via Solaroli 17, 28100 Novara, Italy

**Keywords:** high-fat diet, H3K27 trimethylation, methyltransferase EZH2

## Abstract

Background/Objectives: Methyltransferase EZH2-mediated H3K27me3 is involved in liver inflammation and fibrosis, but its role in hepatic metabolic derangements is not yet clearly defined. We investigated if a high-fat diet (HFD) induced early changes in EZH2 expression and H3K27 me3 in the liver of mice. Methods: Five-week-old mice were fed an HFD or a low-fat diet (Control) for 2 weeks (2 W) or 8 weeks (8 W). Body weight was recorded weekly. Glycemia and oral glucose tolerance were assessed at baseline and after 2 W–8 W. Finally, livers were collected for further analysis. Results: As expected, mice that received 8 W HFD showed an increase in body weight, glycemia, and liver steatosis and an impairment in glucose tolerance; no alterations were observed in 2 W HFD mice. Eight weeks of HFD caused hepatic EZH2 nuclear localization and increased H3 K27me3; surprisingly, the same alterations occurred in 2 W HFD mice livers, even before overweight onset. We demonstrated that selective EZH2 inhibition reduced H3K27me3 and counteracted lipid accumulation in HUH-7 cells upon palmitic acid treatment. Conclusions: In conclusion, we point to EZH2/H3K27me3 as an early epigenetic event occurring in fatty-acid-challenged livers both in vivo and in vitro, thus establishing EZH2 as a potential pharmacological target for metabolic derangements.

## 1. Introduction

A high-fat diet (HFD) triggers multiorgan impairment and is associated with detrimental effects on human health. The liver exerts crucial metabolic functions, including lipid metabolism, and is one of the most affected tissues in the context of metabolic derangements. Intake of diet-derived saturated lipids causes insulin resistance, promotes metabolic-dependent inflammation (referred to as metaflammation), leads to fibrosis, alters expression of lipogenic genes, and determines organ weight gain with significant steatosis development [1,2,3].

Recent studies have increasingly explored the hypothesis that chronic long-term hypercaloric diet supply can affect gene expression programs through epigenetic mechanisms and examined their involvement in the progression of metabolic disorders. Fat accumulation, inflammation, oxidative stress, and fibrosis can indeed be regulated by epigenetic mechanisms, including histone post-translational modifications, DNA methylation, and non-coding RNAs [4,5].

Methyltransferase EZH2 (Enhancer of Zeste Homolog 2), the catalytic subunit of the Polycomb Repressor Complex 2 (PCR2), regulates gene expression via trimethylation of lysine 27 on histone H3 (H3K27me3) [6]. The main function of EZH2 is to suppress gene expression and induce chromatin compaction [7]. Notably, EZH2 overexpression has been mainly described in fibrosis and cancer, showing a positive correlation with their progression [6,8]. Recently, the role of EZH2 besides in cancer has been highlighted in different organs, including the liver [8]. Lee and colleagues demonstrated that EZH2 inhibition reduces liver inflammation and fibrosis in preclinical models of advanced non-alcoholic steatohepatitis [9], resulting in decreased expression of IL-6 (Interleukin-6), IL-1β, Interferon-γ (INF-γ), transforming growth factor β (TGF- β), and connective tissue growth factor (CTGF). However, to date, the epigenetic alterations that occur in the liver in response to HFD have been poorly investigated.

Our study aims to determine whether the EZH2/H3K27me3 axis could represent a very early epigenetic marker of liver impairment upon HFD feeding as an early detection of fat accumulation in the liver. Early intervention strategies could be important to prevent fatty liver damage and to improve hepatic-related outcome. Specifically, we investigate the correlation between the onset of metabolic derangements and H3K27me3 modulation in a preclinical model of diet-induced dysmetabolism (8 weeks of HFD feeding), and subsequently, we try to understand how early this correlation occurs (2 weeks of HFD feeding).

## 2. Materials and Methods

### 2.1. Animal Model

Adult male C57BL/6 J mice of four weeks of age were utilized. Mice, kept at 3–5/cage with access to water and food ad libitum, were housed in a light-controlled (12 h light, 12 h dark) and temperature-controlled (22–24 °C) room in high-efficiency particulate air (HEPA)-filtered Thoren units (Thoren Caging Systems) at the University of Piemonte Orientale animal facility, Novara, Italy. Animal care and handling were performed in accordance with the Italian law on animal care (D.L. 26/2014) as well as the European Directive (2010/63/UE) and approved by the Organismo Preposto al Benessere Animale (OPBA) of University of Piemonte Orientale, Novara, Italy (DB064.61).

#### 2.1.1. Diet Administration

A timeline representation of the experimental design can be found in Appendix A. Thirty-eight animals started receiving an LFD (low-fat diet, 13% kcal from fat, 67% kcal from carbohydrates, 20% kcal from proteins, (Laboratori Piccioni, Gessate, Milan, Italy) instead of the normal chow diet provided by the animal facility from the fourth week of age to get used to a refined diet [10]. After one week (five weeks of age), animals were randomly divided into 4 groups: two groups continued to be fed a low-fat diet (control groups) for eight (*n* = 9) or two (*n* = 9) weeks, while the other two groups of animals were fed with an HFD (60% kcal from fat, 21% kcal from carbohydrates, 19% kcal from proteins, Laboratori Piccioni) (HFD groups) for eight (*n* = 10) or two (*n* = 10) weeks (Appendix A). Animal body weight, food, water, and caloric intake were recorded weekly. At the end of the diet regiments, mice were euthanized, and livers were collected and preserved in an optimal cutting temperature (OCT) compound or snap frozen at −80 °C for morphological and bio-molecular analyses.

#### 2.1.2. Oral Glucose Tolerance Test

The oral glucose tolerance test (OGTT) was performed before the beginning of the experiment and after 2 and 8 weeks of diet supplementation. Glucose (2 g/kg) was administered by oral gavage after a fasting period of 18 h. The concentrations of serum glucose were measured with a conventional glucometer (GlucoMen LX kit, Menarini Diagnostics, Bagno a Ripoli, Florence, Italy) before glucose administration and after 15, 30, 60, and 120 min.

#### 2.1.3. Oil Red O Tissue Staining and Analysis

OCT (optimal cutting temperature)-embedded livers were cryo-sectioned at −26 °C (thickness of 8 μm) using Superfrost Slide (Epredia, Breda, Netherlands) and rinsed in PBS. Oil Red O solution (0.5% in isopropanol, Sigma-Aldrich, Darmstadt, Germany) was used to determine lipid droplets in the liver to a final 3:2 concentration in deionized water. Sections were stained for 15 min and then rinsed in water. Slides were scanned by ZEISS Axioscan 7 (Carl Zeiss Microscopy, Jena, Germany) (setting: REF_BF H&E Slide 40x program). Quantitative analyses were executed using Qu-Path 0.5.0. and ImageJ software 1.8.0_345. At least 50 areas per sample were analyzed. The setting used for analysis was: HUE 0–255; saturation 145–255; brightness 145–255. The stained area was calculated as a percentage of the total liver area analyzed.

### 2.2. Reagents and Antibodies

The polyclonal antibodies specific to EZH2 (at a 1:1000 dilution), histone H3 trimethyl-lysine 27 (H3K27me3, 1:1000), histone H3 (1:2000), and the monoclonal antibody specific to KDM6B (1:1000) were purchased from Active Motif (La Hulpe, Belgium). Anti-rabbit IgG and anti-mouse IgG peroxidase-conjugated antibodies and reagents, including palmitic acid, were from Sigma-Aldrich (St. Louis, MO, USA). The nitrocellulose membrane and ECL were bought from Bio-Rad (Hercules, CA, USA). The EZH2-selective inhibitor, EPZ-6438, was from Selleckchem (Houston, TX, USA).

### 2.3. Cell Cultures

HUH-7 cells, derived from human hepatocellular carcinoma (provided by CLS, Cell Lines Service GmbH, Wageningen, Netherlands), were maintained in DMEM (Sigma-Aldrich, Cat. No. D5671) supplemented with 10% fetal bovine serum (Gibco, ThermoFisher, Waltham, Massachusetts, USA) Cat. No. 10270), 2 mM L-glutamine (Sigma-Aldrich), and 1% penicillin/streptomycin solution (Sigma-Aldrich)). When 80% confluence was reached, cells were plated for experiments. At 24 h after plating, cells were treated with palmitic acid (PA, 100 µM, PA group), PA, and the specific EZH2 inhibitor EPZ-6438 (3.3 µM) (PA + EPZ group). EPZ-6438 concentration was chosen according to previous tests. PA time and concentration were chosen based on relevant scientific literature using models and cells similar to our procedure [11,12,13]. Considering that albumin concentration is essential to determine the concentration of available free fatty acid (FFA), the PA was complexed with bovine serum albumin (BSA; Sigma-Aldrich Cat. No. A9647) at a 4:1 molar ratio considering the albumin concentration already present in the medium due to the FBS supplementation. To warrant the same basal condition, BSA was added in all experimental groups, including the control group (VH).

At 24 h after treatment, cells were lysed for histones extraction as described below, or were fixed in PFA 4% for Oil Red O staining.

### 2.4. Cell and Tissues Fractionation and Immunoblotting

Cell fractionation was carried out starting with 3 × 10^6^ HUH-7 cells. Briefly, cell pellets were resuspended in buffer A (10 mM Tris, 10 mM KCl, 1.5 mM MgCl_2_, 300 mM sucrose, and protease inhibitors) and kept on ice for 10 min. For tissue fractionation, 50 mg of liver tissue from each mouse was homogenized in the same buffer A. After centrifugation at 2000× *g* for 10 min, the supernatant contained membranes and the cytosolic fraction (that was subsequently centrifuged at 10,000× *g* for 60 min). Pellets obtained from the first centrifugation were resuspended in buffer B (50 mM Tris 400 mM NaCl, 1 mM EDTA, and 1% NP40) and incubated on ice for 10 min before centrifugation at 5000× *g* for 5 min. The resulting pellet was again resuspended in buffer B plus protease inhibitors by vortexing, incubated on ice, and centrifuged at 5000× *g* for 5 min. The resulting nuclei were pelleted, and protein extraction was carried out by sonication (2 cycles, 1 s). Lysates were then centrifuged at the highest speed for 15 min at 4 °C, and the supernatant was collected (nuclear fraction). Cytosolic and nuclear extracts were then used for immunoblotting. Protein concentration was determined, and proteins were separated by SDS-PAGE under reducing conditions. Following SDS-PAGE, proteins were transferred to nitrocellulose, reacted with specific antibodies, and then detected with peroxidase-conjugated secondary antibodies and a chemiluminescent ECL reagent. Digital images were taken with the Bio-Rad ChemiDoc™ Touch (Hercules, California, USA) and quantified using Bio-Rad Image Lab 5.2.1.

### 2.5. Oil Red O Cell Staining and Analysis

HUH-7 liver immortalized cells were washed with PBS, stained with a 3:2 Oil Red O solution (0.5% in isopropanol diluted in deionized water; Sigma-Aldrich) for 15 min, and then rinsed.

Slides were scanned with ZEISS Axioscan 7 setting REF_BF H&E Slide 40x program. Quantitative analyses were executed via Qu-Path and ImageJ software. The settings used for analysis were HUE 0–255, saturation 0–255, and brightness 0–185. The stained area was calculated as a percentage of the total area analyzed.

### 2.6. Statistical Analyses

All statistical analyses and data visualizations were performed in GraphPad Prism 8.0.2. For statistical analysis of body weight and OGTT, two-way ANOVA with the Sidak post-hoc test was used. The other results were analyzed with Student’s *t*-test or one-way ANOVA with Tukey’s post-hoc test depending on group number. For all analyses, significance was defined as *p* < 0.05.

## 3. Results

### 3.1. An HFD Affects Mice Body Weight and Glucose Homeostasis after 8 Weeks but Not after 2 Weeks

Mice exposed to the hypercaloric diet showed a significant increase in body weight (Figure 1A) starting from the fifth week of diet administration compared to control mice, resulting a full-blown weight gain at 8 weeks (8 W).

Weight gain progression correlated with fasting glycemia and OGTT, indicating an impairment in glycemic homeostasis at 8 weeks (Figure 1B–D). Notably, mice fed an HFD for only 2 weeks (2 W) did not exhibit impaired glucose control (Figure 1E–G).

### 3.2. An HFD Affects Liver Weight and Steatosis after 8 Weeks but Not after 2 Weeks

Livers from mice were collected and weighted, and after 8 weeks of an HFD, mice exhibited an increase in liver mass compared to the control group (Figure 2A). In contrast, after just 2 weeks of an HFD, no significant changes in liver mass were observed (Figure 2B).

These results correlated with hepatic steatosis. After 8 weeks of the diet regimen, we observed a marked accumulation of lipid droplets in the HFD 8 W group compared to the control 8 W group (Figure 2C,E), as evidenced by the histological analysis, while no effects were observed in the 2 week HFD-fed animals (Figure 2D,F).

### 3.3. Long HFD Exposure Induces an Increase in H3K27me3 Levels Mediated by EZH2

After establishing that 8 weeks of HFD feeding led to increased body and liver weights (Figure 1A and Figure 2A,B) and impaired glucose homeostasis (Figure 1B–D), we investigated the impact of 8 weeks of HFD exposure on the H3K27me3 (trimethylation of lysine 27 of histone 3) epigenetic modification. Livers obtained from the 8 W control group and 8 W HFD mice were collected, and H3K27me3 levels were analyzed. Histone extraction, followed by Western Blot analysis, revealed that the H3K27me3 profile differed between the two diet regimens. In particular, as shown in Figure 3A, we observed a significant increase (*p* < 0.05) in the prevalence of H3K27me3, without any change in the expression of H3, in response to 8 weeks of HFD feeding. We therefore investigated whether an HFD altered liver levels of H3K27me3 by regulating the expression of EZH2. As reported in Figure 3A, we demonstrated that HFD feeding increased EZH2 expression in the nuclear fraction, while cytoplasmic EZH2 fraction (Appendix A) was unaffected by HFD feeding. Surprisingly, we did not detect any change in the expression of H3K27me3-associated demethylase KDM6B (Jumonji domain-containing protein-3) (Appendix A). These data highlight that an in vivo model of HFD feeding is characterized by an intra-hepatic increase in nuclear EZH2, which corresponds to up-regulation of H3K27me3 levels.

### 3.4. Short HFD Exposure Induces an Increase in H3K27me3 Mediated by EZH2

As reported in Figure 1 and Figure 2, we demonstrated that 8 weeks of an HFD caused an increase in body and liver weight and hepatic steatosis in mice, which correlated with an increase in the prevalence of H3K27me3 in the liver (Figure 3A). The persistence of epigenetic alterations has been described as one of the molecular mechanisms that triggers metabolic memory [14]. Therefore, we extended our investigation to an early time-point, specifically after 2 weeks of HFD feeding. As expected, we observed no effects of an HFD for 2 W on body weight, liver mass, or glucose homeostasis (Figure 1). Upon analyzing the levels of H3K27me3 and EZH2 expression, we observed a significant increase (*p* < 0.05) in H3K27me3 in response to short-term HFD feeding (Figure 4A), with no change in the expression of H3. We therefore investigated the expression levels of EZH2. As reported in Figure 4A, 2 weeks of HFD feeding resulted in increased EZH2 levels in the nuclear fraction, while the cytoplasmic fraction remained unchanged (Appendix A).

The reported results indicate that both the upregulation of H3K27me3 and EZH2 occurs in response to HFD exposure, independent of the duration of feeding.

### 3.5. EPZ-6438, a Selective EZH2 Inhibitor, Reduces H3K27me3 Levels and Counteracts Lipid Accumulation in HUH-7 Cells Treated with Palmitic Acid

To investigate whether pharmacological modulation of H3K27me3 could affect liver lipid accumulation, we established a well-described in vitro model that mimics lipotoxicity induced by free fatty acids. We treated the HUH-7 cells, established from male hepatoma tissue, with palmitic acid (PA) at a concentration of 100 μM for 24 h in the presence or absence of the EZH2 inhibitor EPZ-6438 at 3.3 μM. We investigated whether PA treatment influenced the levels of H3K27me3 and EZH2 expression in both cytoplasmic and nuclear fractions.

As reported in Figure 5A, we demonstrated that PA treatment significantly increased the levels of H3K27me3 and nuclear EZH2, consistent with our findings in mice subjected to HFD feeding. The combined treatment of PA and EPZ-6438 significantly counteracted the induction of H3K27me3 exerted by PA (Figure 5A). Furthermore, we report that PA treatment led to significant accumulation of lipid droplets in HUH-7 cells, as assessed by Oil Red O Staining. Lipid accumulation was partially reversed by the combined treatment of PA and EPZ-6438, as shown in Figure 5D–E. These data clearly demonstrate that the expression and activation of EZH2 play a direct role in lipid accumulation.

## 4. Discussion

Overweight and obesity are significant risk factors for lipid accumulation in the liver. Excessive hepatic fat accumulation can lead to inflammation, liver cell injury, and the progression to more severe conditions such as non-alcoholic steatohepatitis (NASH) and, eventually, cirrhosis [15,16]. Extensive research indicates that the intricate interplay of multiple factors, including insulin resistance, inflammation, and oxidative stress, contributes to the pathogenesis of hepatic lipid accumulation in individuals who are overweight or obese. Of note, the initial detrimental mechanisms leading to fat accumulation in the liver begin to onset even at early stages of metabolic derangement and even before the onset of overweight [17]. The factors involved in the early/late onset of liver-associated metabolic derangement have not yet been fully investigated, but a substantial body of evidence suggests that epigenetic mechanisms could play a pivotal role.

Among epigenetic players, EZH2, through H3K27me3, is known to exert a crucial role in the regulation of inflammatory and fibrotic markers in liver parenchyma [9,18]. Data reported by Mann and colleagues revealed that the expression of EZH2 was up regulated in activated hepatic stellate cells (HSCs) compared with quiescent HSCs [19]. Emerging evidence indicates HSCs exert a pivotal role in hepatic fibrosis [19], a typical consequence of chronic diet-induced dysmetabolism [20]. EZH2 may drive hepatic fibrosis via H3K27me3 in order to repress anti-fibrotic gene expression [21]. Among those under H3K27me3 control, small mother against decapentaplegic 7 (Smad7) and Dickkopf-related protein 1 (Dkk1) are two anti-fibrotic genes known for their ability to inhibit pro-fibrotic signaling mediated by TGF-β [22,23] and Wnt [24,25], respectively. Treatment with an EZH2 inhibitor, indeed, could prevent the trans differentiation of HSCs to highly proliferative and fibrogenic myofibroblast-like cells (MFBs) [26]. Some authors reported that EZH2 silencing restored Dkk1 expression, preventing TGF-β-induced proliferation of HSCs and expression of alpha smooth muscle Actin (α-SMA) [26].

In line with these data, Rosa Martin-Mateos and colleagues demonstrated that EZH2 is strongly involved in TGF-β-mediated HSC activation. Inhibition of EZH2 attenuates fibrogenic gene transcription in TGF-β-mediated HSC activation in in vitro and in vivo liver fibrosis murine models, reducing the expression of fibronectin, α-SMA, and collagen 1α1 [27]. Additionally, the EZH2 inhibitor promoted the expression of the activin membrane-bound inhibitor (BAMBI); IL-10 (interleukin-10); and cell cycle regulators, including cyclin-dependent kinase inhibitor 1A (Cdkn1a) as well as growth arrest and DNA damage-inducible alpha (Gadd45a) and beta (Gadd45b), leading to TGF-β/Smads suppression and an anti-inflammatory response [26]. Additionally, another anti-fibrotic gene repressed by EZH2 is peroxisome proliferator-activated receptor gamma (PPAR-γ) [28], a ligand-activated transcription factor that promotes protective effects against including liver fibrosis [29].

In this study, we demonstrate an increase in EZH2 expression and nuclear localization as well as increased H3K27me3 levels in the livers of overweight mice that were fed an HFD for 8 weeks. Moreover, we did not observe any changes in the levels of the H3K27me3-specific demethylase KDM6B. H3K27me3 is known for its long-lasting effects on transcriptional regulation in conditions of diet-induced metabolic derangement. In this context, Blin and colleagues demonstrated that chromatin-repressive marks such as H3K27me3 represent a potential candidate in the understanding of the long-term “developmental programming” effects induced by early exposure to nutritional excess [30]. Thereby, EZH2-mediated H3K27me3 can be considered among the players involved in the so-called “metabolic memory”. The hypothesis of metabolic memory suggests that uncontrolled glycemic/weight control in early stages of metabolic derangement can have a long-lasting effect on increasing the risk of complication onset later in life [31]. The mechanisms involved in metabolic memory have been so far poorly investigated; epigenetic modifications, and particularly H3K27me3, could represent a promising area for further investigation in this context.

To understand how early H3K27me3 occurs, we continued our experimental protocol analyzing the livers of mice fed an HFD for only 2 weeks (short HFD group), which corresponds to a time that precedes the onset of overweight and the impairment of glucose homeostasis. Surprisingly, we detected, at this early time-point, a significant upregulation of the nuclear levels of EZH2 and H3K27me3, similar to what we observed in overweight mice (HFD for 8 weeks). Here we reveal, for the first time, an early epigenetic alteration resulting from a short HFD challenge occurring prior to the onset of a full-blown overweight/glucose impairment. These findings contribute to consolidating the hypothesis concerning the involvement of EZH2/H3K27me3 in metabolic memory. To strengthen this idea, we plan to perform a new in vivo experimental protocol consisting of a short HFD priming of 2 weeks (similar to the current study), followed by a recovery phase on a control diet, and then a new HFD challenge.

To properly correlate the hepatic modulation of EZH2 expression upon short/long HFD consumption, we established a well-described in vitro model of non-alcoholic fatty liver disease (NAFLD) using immortalized HUH-7 cells challenged with PA (24 h, 100 µM) [32] and EPZ-6438 (24 h, 3.3 µM).

Firstly, we demonstrated that the results obtained from the in vitro model after PA challenge mimicked those obtained in vivo after an HFD. Specifically, we observed an increase in the expression of EZH2/H3K27me3 in cells that received PA compared to vehicle-treated cells. As a readout for hepatic metabolic condition, we measured liver weight in vivo, while we evaluated the accumulation of intracellular lipid droplets in vitro. As predicted, PA treatment induced an increase in fat droplets.

EPZ-6438 treatment reduced H3K27me3 levels and nuclear expression/localization of EZH2 induced by PA. More interestingly, it significantly counteracted PA-dependent lipid accumulation.

These results clearly demonstrate that EZH2 serves as a direct regulator of fat accumulation. Very recently, EZH2 has been proposed as a potential therapeutic target for advanced NAFLD [9]. Our results reveal that EZH2 deregulation is an early indicator of metabolic derangement. For this reason, we believe that EZH2-targeted therapy could represent an attractive early approach to mitigate the late risk of liver injury. Currently, EPZ-6438 (tazemetostat) is a first-in-class U.S. FDA-approved EZH2 inhibitor for follicular lymphoma and epithelioid sarcoma [33,34,35].

Even in the context of drug repurposing, it will be strategic to perform an in vivo pharmacological study with EPZ-6438 in the setting of both short and long HFD feeding. Finally, our next goal to overcome the limitation of this study will be to investigate the gene sets regulated by H3K27me3 through RNA sequencing followed by CHIP (chromatin immune precipitation) analysis with the aim of identifying more specific druggable molecular pathways.

## 5. Conclusions

In conclusion, we identify EZH2-mediated H3K27me3 as an early epigenetic change occurring in livers challenged by fatty acids, both in vivo and in vitro. Our finding supports the hypothesis that EZH2 represents a potential pharmacological target for addressing metabolic disorders.

## Figures and Tables

**Figure 1 nutrients-16-03260-f001:**
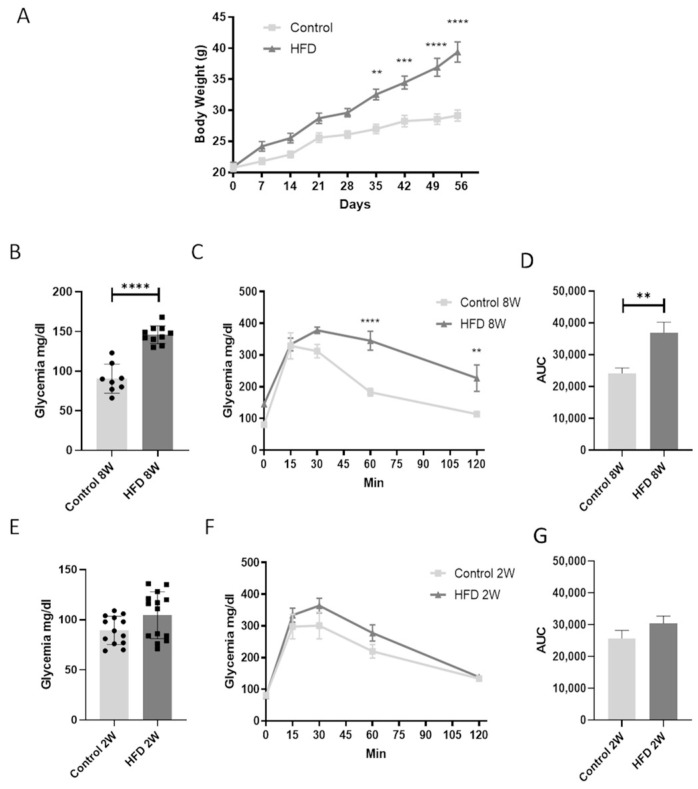
An HFD for 8 W but not 2 W affects mice body weight and glucose homeostasis. (**A**). Mice body weight, glycemia, oral glucose tolerance test, and relative area under the curve (AUC) of mice fed for 8 weeks (8 W) with a control diet or HFD (**B**–**D**). Glycemia, oral glucose tolerance test, and relative area under the curve (AUC) of mice fed for 2 weeks (2 W) with a control diet or a HFD (**E**–**G**). ** = *p* < 0.01 vs. control, *** = *p* < 0.001 vs. control, **** = *p* < 0.0001 vs. control. Data are presented as mean ± SEM, *n* = 8–14 per group.

**Figure 2 nutrients-16-03260-f002:**
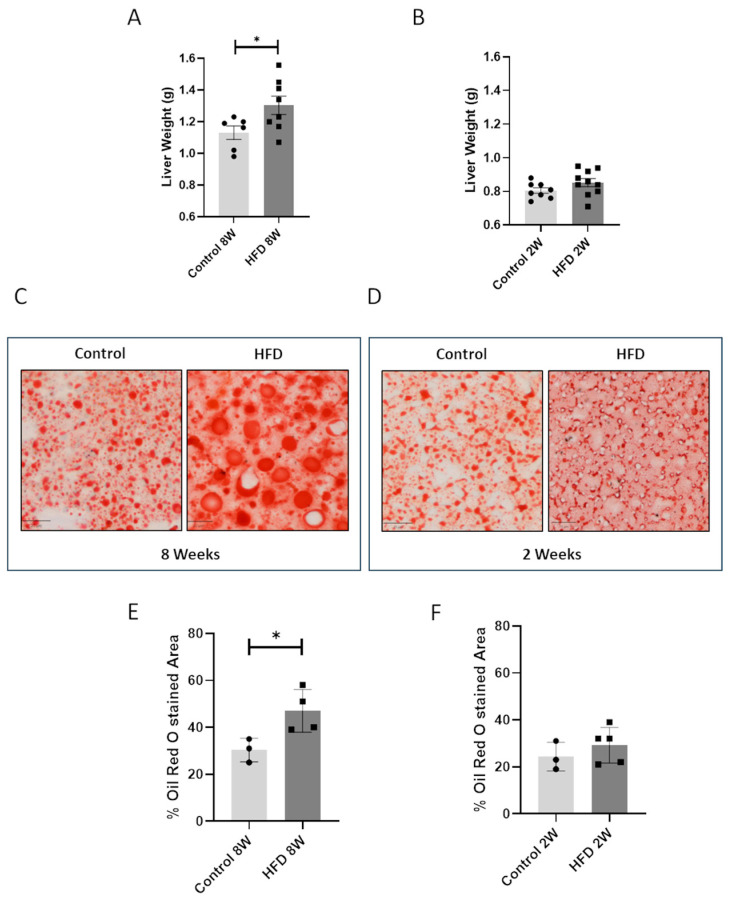
An HFD at 8 W but not 2 W affects mice liver steatosis. Liver weight of mice fed a control diet or an HFD for 8 weeks (**A**) or 2 weeks (**B**). * = *p* < 0.05. Data are presented as mean ± SEM, *n* = 6–10 per group. Representative images and quantitative analysis of Oil Red O Staining on livers of mice fed a control diet or an HFD for 8 (**C**,**E**) or 2 weeks (**D**,**F**). Line represents a length of 20 µm (**C**,**D**). * = *p* < 0.05. Data are presented as mean ± SEM, *n* = 3–4 per group.

**Figure 3 nutrients-16-03260-f003:**
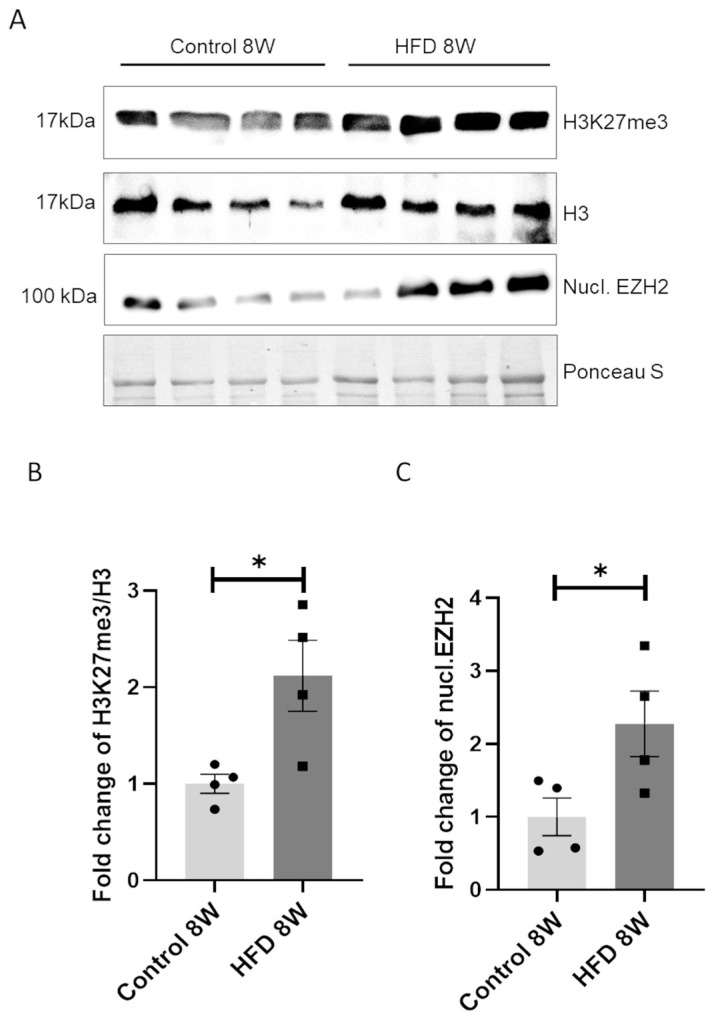
An HFD for 8 W increased H3K27me3 levels and nuclear localization of EZH2. (**A**) Representative Western blotting analysis of nuclear autoradiogram of EZH2 (nucl. EZH2), H3K27me3, and H3 in four mouse liver samples per each group (control and HFD), normalized to total protein (Ponceau S staining). Fold change in (**B**) H3K27me3 levels and (**C**) nuclear EZH2 (nucl. EZH2). * = *p* < 0.05. Data are presented as mean ± SEM, *n* = 4 per group.

**Figure 4 nutrients-16-03260-f004:**
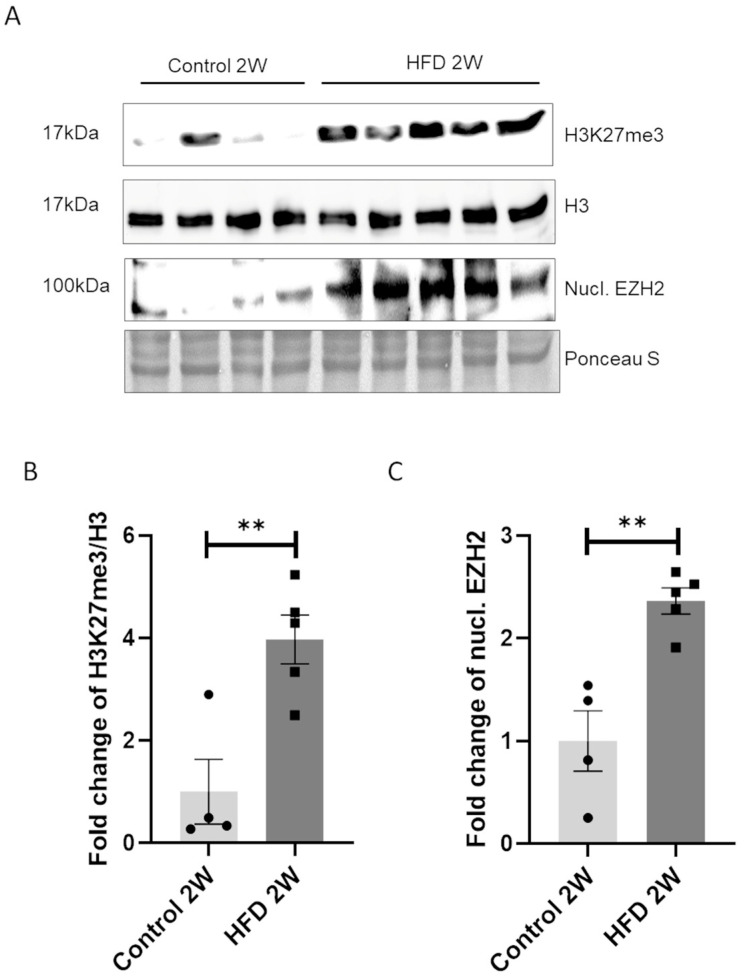
An HFD for 2 W increases H3K27me3 levels and expression and nuclear localization of EZH2. (**A**) Representative Western blot analysis of nuclear level of EZH2 (nucl. EZH2), H3K27me3, and H3 in four mouse liver samples per control group and five per HFD 2 W group, normalized to total protein (Ponceau S staining). Fold change in (**B**) H3K27me3 levels and (**C**) nuclear EZH2 (nucl. EZH2). ** = *p* < 0.01. Data are presented as mean ± SEM, *n* = 4/5 per group.

**Figure 5 nutrients-16-03260-f005:**
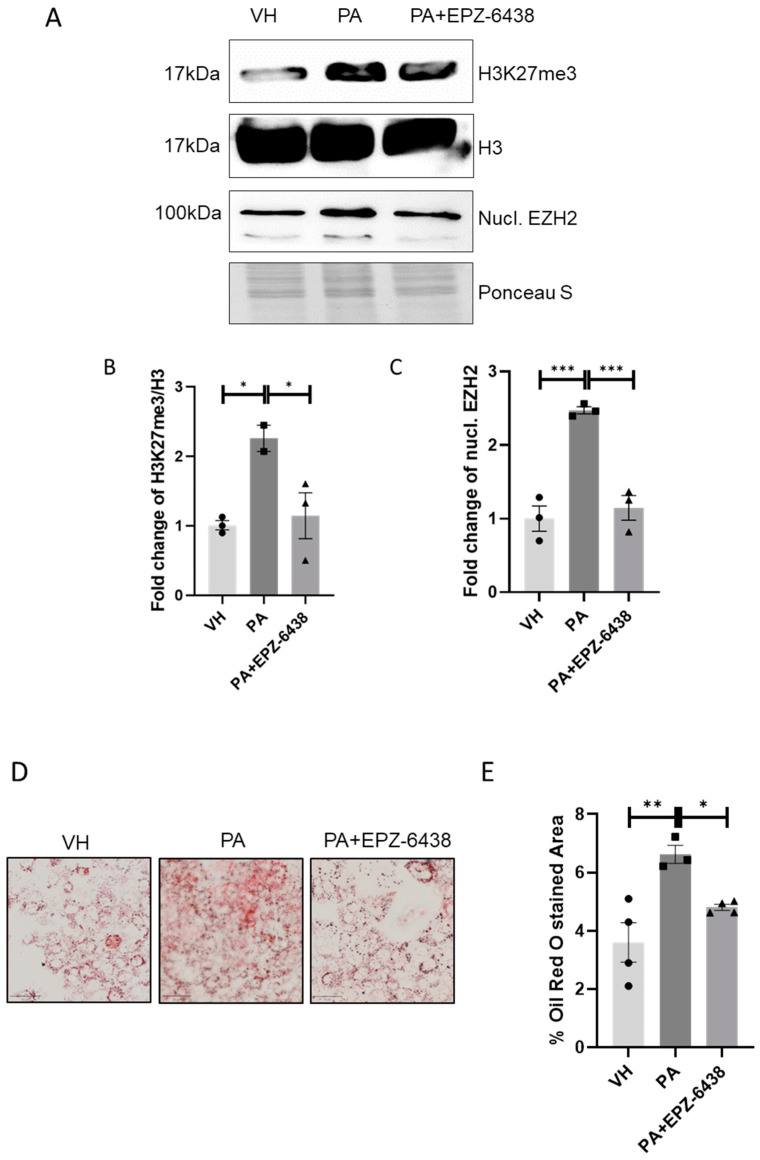
EPZ-6438 treatment reduces H3K27me3 levels and counteracts lipid accumulation induced by palmitic acid in HUH-7 cells. (**A**) Representative Western blot analysis of nuclear EZH2 (nucl. EZH2), H3K27me3, and H3 in HUH cells treated for 24 h with 100 µM palmitic acid in combination or not with EPZ-6438 3.3 µM. Fold change of (**B**) H3K27me3 levels and (**C**) nuclear EZH2 (nucl. EZH2). Representative images and quantitative analysis of Oil Red O staining on HUH-7 cells treated with PA with or without EPZ-6438 (**D**,**E**). Line represents a length of 50 µm (**D**). * *p* < 0.05, ** *p* < 0.01, *** *p* < 0.001. Data are presented as mean ± SEM, *n* = 3–4 experiments.

## Data Availability

The raw data supporting the conclusions of this article will be made available by the authors on request.

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
