# Peer review of "EZH2-Mediated H3K27 Trimethylation in the Liver of Mice Is an Early Epigenetic Event Induced by High-Fat Diet Exposure"

_nutrients, 2024, doi:10.3390/nu16193260_

Round 1

Reviewer 1 Report

Comments and Suggestions for Authors

The topic investigated in the manuscript by Pinton G. et al. is interesting and relevant. The authors have chosen a timely subject with the potential to make a meaningful contribution to the field. While the findings are promising, the study would benefit from major editing and, if possible, additional experimental work to strengthen the conclusions.

In this study, the authors report that after 8 weeks of high-fat diet (HFD) feeding, mice developed weight gain, insulin resistance, and liver steatosis—a phenotype widely described in previous studies. The novelty, however, is that this was accompanied by an increase in the levels of histone methyltransferase EZH2 and its corresponding epigenetic mark (H3K27me3), with no observed changes in the levels of the histone demethylase. However, in a 2-week feeding experiment, the authors observed only an increase in histone methyltransferase, the epigenetic mark, and no other changes. The authors propose that the increase in H3K27me3 could be an early indicator of hepatic fat accumulation. Unfortunately, the authors did not measure the levels of specific H3K27me3 targets to establish a causal link between the increase of the epigenetic tag and the development of steatosis. Following this, the authors conducted an in vitro study using hepatic cancer cells challenged with palmitic acid, which resulted in fat accumulation that was prevented by an EZH2 inhibitor already FDA-approved for human use. I was puzzled by the choice to conduct this part of the study in vitro. Why didn't the authors perform this experiment in vivo? Doing so would have greatly enhanced the study, as they acknowledge in the discussion.

Additionally, what was the content of palmitic acid in the HFD? The diet suppliers typically provide the fatty acid profiles of their formulas upon request, which would be valuable information to include. How was the incubation time and palmitic acid concentration selected in the cell study?

Why is there a discrepancy between the number of mice in each dietary group (n=9) and the data presented in Figures 3, 4, and S2 (n=4-5)? 

Please review the use of abbreviations throughout the manuscript. For instance, LFD (mentioned in section 2.1) and FFA (mentioned in section 2.3) are not explained. Additionally, abbreviations such as EZH2 and H3K27me3 are introduced in the Introduction and reintroduced in the Discussion, which is redundant.

In the Methods section, the dilution of the antibodies should be specified. Please also include the components of the experimental diets and the typical fatty acid profile.

The figures would also benefit from some improvement. I also propose adding a new graph representing the Area Under the Curve (AUC) for the glycemia in the OGTTs to provide a clearer visualization of the glucose metabolism data. The figure captions in Figures S1 and S2 should be provided. I would suggest rearranging the figures for clarity, such as combining Figures 4B and 4C in one figure and presenting the data in chronological order (2 weeks and 8 weeks). Additionally, I recommend replacing the term "SD" with "control" throughout the manuscript for clarity. Fig 5 legend: … n=3 (Fig 5E)? Figure S1A requires a major improvement. For example, why are weeks 4 and 6 represented if no experimental work was done at these time points? Add n, age, and gender of the mice. Include the OGTTs performed, as well as the liver collection and studies. Replace SD with Control.  Update the table to show 4.3 instead of 4,278 and 5.7 instead of 5,736, ensuring units are not repeated. Move Figures S1 and S2 to the main body of the manuscript for better integration.

Lastly, the discussion could be enhanced by incorporating examples of previous studies that support the observed HFD phenotype and by exploring potential H3K27me3 targets relevant to the liver phenotype. Additionally, a thorough review of the English language usage would improve the readability and clarity of the manuscript.

Comments on the Quality of English Language

A careful review of the English usage would enhance the readability and clarity of the manuscript.

Reviewer 2 Report

Comments and Suggestions for Authors

Recent in vitro and in vivo studies have revealed the epigenetic role of EZH2 in human diseases, including NAFLD and cancer, but the involved pathogenic mechanisms are still, poorly investigated understood. In the current original study the authors show for the first time, the early involvement of EZH2/H3K27me3 of an epigenetic alteration occurring in metabolic memory, after a HFD challenge and the EZH2 deregulation as an early indicator of this metabolic derangements. The study is interesting, well written and organized, and fits in the scope of the journal. The abstract shows the key points and the introduction justifies the aim of the study; methods are clearly described and statistical evaluation is correct. The authors have used in vivo and in vitro experiments in order to validate the role of EZH2 in NAFLD; data are analyzed in the discussion section in the context of existing literature. Although the study’s novelty, significance, contribution, and possible impact on the field are highlighted the need for further research is also emphasized. The authors set further goals, in order to elucidate the detailed actions of EZH2/H3K27me3 in the pathogenesis and progression of NAFLD in the concept of metabolic memory with ultimate goal the management/reverse of the disease with newly developed EZH2 inhibitors.  

Minor comments

Since there are several newly developped EZH2 inhibitors, the authors should  justify their selection.

Are there any limitations of the study? 

Comments on the Quality of English Language

Recent in vitro and in vivo studies have revealed the epigenetic role of EZH2 in human diseases, including NAFLD and cancer, but the involved pathogenic mechanisms are still, poorly investigated understood. In the current original study the authors show for the first time, the early involvement of EZH2/H3K27me3 of an epigenetic alteration occurring in metabolic memory, after a HFD challenge and the EZH2 deregulation as an early indicator of this metabolic derangements. The study is interesting, well written and organized, and fits in the scope of the journal. The abstract shows the key points and the introduction justifies the aim of the study; methods are clearly described and statistical evaluation is correct. The authors have used in vivo and in vitro experiments in order to validate the role of EZH2 in NAFLD; data are analyzed in the discussion section in the context of existing literature. Although the study’s novelty, significance, contribution, and possible impact on the field are highlighted the need for further research is also emphasized. The authors set further goals, in order to elucidate the detailed actions of EZH2/H3K27me3 in the pathogenesis and progression of NAFLD in the concept of metabolic memory with ultimate goal the management/reverse of the disease with newly developed EZH2 inhibitors.  

Minor comments

Since there are several newly developped EZH2 inhibitors, the authors should  justify their selection.

Are there any limitations of the study? 

Round 2

Reviewer 1 Report

Comments and Suggestions for Authors

Thank you for addressing my comments. Your responses were satisfactory overall, and your work is integral to our understanding of this field. However, the discussion still needs some refinement.

To clarify my earlier point, H3K27me3 is a repressive epigenetic mark, meaning that an increase in this mark leads to reduced gene expression. The discussion would still benefit from referencing studies demonstrating how increased H3K27me3 levels cause the downregulation of specific genes relevant to the phenotype observed in your mice, explaining its relevance to provide a more precise context for the reader.

The figures and their legends still appear to lack attention to detail. For instance, in Figure 1, the abbreviation "2W" is not explained. To enhance clarity and understanding, please ensure that all abbreviations and symbols used in the figures are clearly defined in the legends. 

Additional minor points:

• Include in your paper an explanation of how the concentration of palmitic acid was chosen, and cite the studies mentioned in your response to the reviewer.

• About the chronological order of data presentation: clearly explain your rationale for this order in the paper to help readers better follow the data.

• New Figure S1 requires further revision. Ensure that time points are presented consistently (e.g., week 0 and 2 weeks). Introduce the abbreviation OGTT. Also, center the text between the two mice to improve visual clarity.

•  Line 159: correct oGTT

•  On line 160, I suggest this text: Notably, mice fed an HFD for only 2 weeks did not exhibit impaired glucose control (Fig. 1E-G).
